# Non-Invasive Microwave-Based Imaging System for Early Detection of Breast Tumours

**DOI:** 10.3390/bios12090752

**Published:** 2022-09-12

**Authors:** Carolina Blanco-Angulo, Andrea Martínez-Lozano, Roberto Gutiérrez-Mazón, Carlos G. Juan, Héctor García-Martínez, Julia Arias-Rodríguez, José M. Sabater-Navarro, Ernesto Ávila-Navarro

**Affiliations:** 1Department of Materials Science, Optics and Electronic Technology, Miguel Hernández University of Elche, 03202 Elche, Spain; 2Department of Communications Engineering, Miguel Hernández University of Elche, 03202 Elche, Spain; 3Neuroengineering Biomedical Research Group, Institute of Bioengineering, Miguel Hernández University of Elche, 03202 Elche, Spain; 4Medical Robotics Research Group, University of Málaga, 29071 Málaga, Spain

**Keywords:** breast phantom, medical imaging, microwave-based measurement, non-invasive measurement, Radio-Frequency antenna system, TRITON X-100, tumour detection

## Abstract

This work introduces a microwave-based system able to detect tumours in breast phantoms in a non-invasive way. The data acquisition system is composed of a hardware system which involves high-frequency components (antennas, switches and cables), a microcontroller, a vector network analyser used as measurement instrument and a computer devoted to the control and automation of the operation of the system. Concerning the software system, the computer runs a Python script which is in charge of mastering and automatising all the required stages for the data acquisition, from initialisation of the hardware system to performing and saving the measurements. We also report on the design of the high-performance broadband antenna used to carry out the measurements, as well as on the algorithm employed to build the final medical images, based on an adapted version of the so-called Improved Delay-and-Sum (IDAS) algorithm improved by a Hamming window filter and averaging preprocessing. The calibration and start-up of the system are also described. The experimental validation includes the use of different tumour models with different dielectric properties inside the breast phantom. The results show promising tumour detection capabilities, even when there is low dielectric contrast between the tumoural and healthy tissues, as is the usual case for dense breasts in young women.

## 1. Introduction

Breast cancer is the most-diagnosed cancer among the female population, making up 11.6% of all female cancers in 2018, according to the WHO’s last report [1]. Accordingly, during the last several years there has been a considerable increase in the research activity on technologies aimed at breast cancer detection [2].

Currently, X-ray mammography is the most widely used medical image method for breast cancer screening and detection. However, this technique shows several drawbacks, such as the need for uncomfortable and painful breast compression, the considerable amount of false-positive tests, especially in young patients with dense breasts, and the use of ionising waves [3,4]. Other current techniques also show significant drawbacks, such as the elevated cost of magnetic resonance imaging (MRI) which hinders its use for early detection purposes, or the need for the patient to take radioactive compounds for the proper generation of the image with positron emission tomography (PET). These disadvantages have motivated the scientific community to pursue new breast cancer detection methods based upon non-invasive, non-ionising, cost-effective techniques. The use of some novel technologies for such an aim has been under study during the last years, such as PhotoAcoustic Tomography (PAT), Diffuse Optical Tomography (DOT) or MicroWave Imaging (MWI). These techniques show the common advantages of being innocuous and not needing to compress the patient’s breast (as happens with X-ray mammography), thereby enhancing the patient’s experience and comfort. 

PAT-based techniques make use of short-pulsed light beams to illuminate the objects under study while the photoacoustic waves, excited by the thermoelastic expansion of the tissues induced by optical absorption, are recorded by wideband ultrasound transducers. Although featuring good spatial resolution and image depths, PAT techniques have the disadvantage of only being useful for the detection of targets of certain sizes, and the complexity of determining the tissue scattering coefficient poses a remarkable limitation in the specific case of breast tumour detection [5]. Concerning DOT techniques, they are based on creating concentration maps of the main intrinsic absorbents in the breast, as some of them are believed to be related to the existence of breast tumours [6]. These methods have the limitation of low spatial resolution, which hinders the detection of early-stage tumours, and relatively low data-processing and image-building speed [7].

MWI techniques are based on the detection of the scattering and absorption of electromagnetic waves at microwave frequencies while travelling through the different tissues in the breast, with a special focus on the remarkable difference found between healthy tissues and tumour tissues. In addition to the common advantages mentioned above, MWI methods show other benefits such as appropriate spatial resolution and image depth for breast tumour detection, the possibility of early detection and tumour detection in dense breasts (especially needed for young patients), relatively low cost and reduced measurement-processing and image-generation times. In this sense, medical imaging systems based on signals with microwave-range frequencies are rising as a promising technique for on-time detection of this kind of breast cancer [4,8]. Moreover, microwave-based images can be used for the detection and characterisation of biological materials [9], thus providing interesting additional information.

The specific properties of electromagnetic wave propagation at microwave frequencies have been demonstrated to be of interest for biomedical applications [10,11]. This range of frequencies covers the most relevant relaxation processes of watery and biological systems [12], allowing for detection, characterisation and tracking of a plethora of phenomena. In addition, the penetration depths attainable with microwave radiation into biological tissues, ranging from a few hundreds of microns to some centimetres [13], render these techniques suitable for many biomedical applications if the proper frequencies and configurations are selected. All these features, in addition to the dependence of the propagation of microwaves upon the permittivity of the tissues, which changes from healthy to malignant [14], as well as the non-invasiveness and non-ionising capabilities [15], have led to the consideration of microwave techniques for medical imaging contexts (e.g., [16,17,18]).

Considering the usual microwave medical imaging systems, they are generally based either on microwave tomography techniques or on radar techniques. When an image is to be generated with microwave tomography systems, the obtained dispersion signals are inverted in order to create a conductivity and permittivity map for the materials the microwaves travel through [4]. Conversely, radar-based systems make use of the return waves coming from the reflections of the different objects to build the images [4]. The operating frequency range for these systems is usually found between 1 and 10 GHz, which supposes a good trade-off between acceptable precision and enough energy penetrating into the biological tissue [3]. Whereas tomography techniques can achieve remarkable precision, the image-generation processing usually requires considerable time, even dozens of minutes [19,20]. Radar techniques, conversely, can produce the images in a faster manner with similar precision [8], although they require sophisticated calibration techniques [4], usually made with in-lab measurements of biological phantoms.

Reliable biological tissue phantoms and models are required so that these systems can be tested, calibrated and set up properly, both for the different parameters of the hardware and the software. To that end, phantoms showing the same electrical parameters (such as the dielectric constant) as each of the actual tissues in a human breast are produced [21]. There is a noticeable difference between the response to microwave propagation in healthy and malignant tissue, mainly due to the elevated water amount contained in tumours, which allows for their microwave-based detection [22]. Reliable phantoms must account for this difference. However, this difference is reduced for denser breasts with lower fat content, as well as for early-stage tumours, which hinders the detection [23,24]. This effect is especially important for patients under 40 years old with dense breasts, for whom the mammography becomes less effective [21]. In this sense, this phenomenon should be addressed by novel techniques and phantoms.

This work has the purpose of starting up a microwave medical imaging system based on radar technology. The system is composed of 16 twin Vivaldi-like antennas which are mastered through a switching network made with five high-frequency switches, as well as a control and feeding subsystem and a computer used to automate the data-acquisition and image-generation processes. The whole system is aimed at the detection and location of tumour phantoms emulating breast cancer cases. As a novelty, in addition to providing a non-ionising fast-imaging system for early detection of breast cancer, we investigate the detection limits of the system by employing different tumour phantoms with reduced dielectric contrast with respect to the breast tissues, thus emulating the dense-breast cancer detection issue.

The rest of the article is organised as it follows: Section 2 explains the hardware and software systems developed to perform the medical imaging measurements; Section 3 discusses the calibration, start-up and fine-tuning of the system for reliable measurements; Section 4 presents the material employed for achieving the biological phantoms and shows the measurements carried out with the proposed system and realistic breast models, as well as the results obtained; finally, Section 5 summarises the main conclusions of the study.

## 2. Microwave System for Early Detection of Breast Tumours

A measuring system has been implemented to carry out measurements of the breast phantoms made with biocompatible material. The system is composed of 16 antennas which are sequentially connected to the Vector Network Analyser (VNA), ZNLE6 from Rohde & Schwarz, Munich (Germany), by means of an electronic switching network. The system is depicted in Figure 1, where a picture of the positions of the antennas and the measurement area is also shown. The measurements are made following the radar principles. The system activates each antenna in order (only one antenna is active at a time), and the active antenna emits a broadband pulse. The pulse is reflected from the different materials making up the objects or bodies under analysis, and the reflected signal is finally received (and recorded) by the active antenna, the same one that emitted it.

The communication with the antennas is handled by a switching network based on commercially available 4-output switches (ZSWA4-63DR+ from Mini-Circuits). These devices provide for high-speed (320 ns), low-loss switching, matched to 50 Ω in the frequency band ranging from 1 MHz to 6 GHz. This switching network is in charge of sequentially connecting each antenna to the VNA. The VNA is configured to work between 10 MHz and 6 GHz, making measurements with 5001 points. All the measurements are always performed with the same port in the VNA. In order to control and select which antenna is connected to the VNA at each time, a subsystem based on a microcontroller (AT91SAM3X8E from Arduino, Ivrea (Italy)) has been included. This control system stores and sweeps a truth table containing the control inputs for the switches according to the antenna activation sequence. Both subsystems (control and switching) are integrated in the bottom part of the system, far enough from the antennas so that the metallic parts cause no interference in the measurement. It should be noted that having only one antenna transmitting and receiving at the same time allows for the use of only 5 switches and fast-processing algorithms. A more precise system could be achieved by having all the antennas receiving the transmitted pulse at the same time, but the considerable increase in the number of required switches, cost, complexity and processing times may entail the loss of the main benefits of radar systems (moderate cost, fast processing, etc.).

Moreover, a software script in the Python language, running on a master computer, has been developed to operate the VNA (through LAN connection) and manage the microcontroller (through a series port connection via USB port) in parallel. This allows us to handle and save the response of each antenna in an independent manner. The Python script is in charge of the VNA initialisation and configuration. After that, the antennas are sequentially selected by means of the microcontroller and the switching network, in order from antenna #1 to antenna #16. At the same time, the right commands are sent to the VNA so that the corresponding calibration is loaded and a measurement of the reflection parameter for each antenna |*S_m_*| is carried out and saved with the selected format. Finally, all the measurement files are loaded into the corresponding directory in the master computer. The measurements are then ready to be processed so that the corresponding images can be produced.

### 2.1. Broadband Printed Antenna

The key elements of the measuring system are the 16 twin broadband antennas designed. They have been expressly designed to achieve a considerable directivity. This improves the detection capabilities, especially when the properties of the target are not greatly differentiated from those of the surroundings, as is the case of tumours in dense breasts. They have been placed on an equally spaced basis along a circumference around the breast phantom under measurement, with an angular separation between every two consecutive antennas of 22.5°. The total number of antennas, and therefore their angular separation, has been set as a trade-off between the largest possible number of antennas to achieve better spatial accuracy in the results and the minimum separation required between the antennas so that the metallic parts of the neighbour antennas do not significantly modify the radiation properties of the active antenna.

The antennas have been designed as a modified version of the common Vivaldi printed antennas, aiming at a reduced size. Vivaldi-like antennas are one of the most-used broadband antenna designs in the literature [25,26]. Their main advantages are having a slightly directive radiation pattern in the aperture direction and showing a logarithmic profile which allows for a considerably broad bandwidth (only limited by the matching in the antenna feed). Conversely, their chief drawback is their big size. This size indeed depends on the dimensions of the aperture of the antenna, which is adjusted to half a wavelength for the lowest design frequency [27]. To overcome this disadvantage while maintaining the excellent radiation features of standard Vivaldi antennas, in this work we use a Vivaldi-like antenna with a modified geometry that allows us to remarkably reduce its size. The antenna has been designed and simulated by means of HFSS (High-Frequency Structure Simulator), v. 2021 R2 from Ansys, Canonsburg (PA, USA).

The design of the antenna is shown in Figure 2. The antenna is printed on both faces of the dielectric substrate, and the classical aperture of the exponential profile in the Vivaldi antennas has been truncated with the purpose of reducing the final size. In addition, three identical director elements have been included in the aperture of the antenna in order to increment the directivity and focus the radiation towards the breast phantom. The antenna is fed thanks to a microstrip feeding line in the bottom face of the substrate. This line implements a radial stub and a triangular taper to enhance the matching and to provide 50 Ω throughout the whole operating frequency range.

The design and optimisation of the antenna was performed with HFSS software by running parametric simulations of the different dimensions, assessing in each case the simulated bandwidth (computed from the reflection parameter S_11_) and directivity. One design goal was to achieve a broad enough bandwidth within the operating range for the rest of the elements involved in the microwave system (maximum frequency of 6 GHz). Another design goal was to obtain a small enough size so that 16 antennas can be properly included in the system, which presents a limit for the minimum operating frequency in the system. After the optimisation process, the final design has a size of 70 × 68 mm^2^ (more than 4× smaller than the size of the equivalent standard Vivaldi antenna [28]), with a bandwidth greater than 3.5 GHz. The final dimensions are summarised in Table 1.

The antennas have been manufactured with low-cost, double-sided copper-clad 1.52 mm-thick FR4 substrate (dielectric constant of 4.4, loss tangent of 0.02). The manufactured antennas are shown in Figure 3. It can be seen that two holes have been made in each antenna so that it can be fixed onto the right position by nylon bolt/nut pairs. An ad hoc plastic piece has been designed and 3-D-printed to hold the antenna and properly fix its position, to which the antenna is attached by the bolt/nut pairs. We have verified that neither the nylon bolt/nut pairs nor the plastic piece have significant influence on the response of the antennas.

The responses of the fabricated antennas have been characterised inside an anechoic chamber with the VNA. Figure 4 plots the results of the S_11_ parameter for one of the antennas (identical responses have been obtained for them all), both simulated and measured. As can be observed, a good agreement is obtained, which validates the design and optimisation process carried out. The experimental operating frequency range of the antenna is from 1.2 GHz to 5 GHz, which means 123% bandwidth for a central frequency of 3.1 GHz.

The E-plane and H-plane radiation patterns, both simulated and measured, are shown in Figure 5. For a better understanding of the results, due to the large number of frequencies involved, the diagrams have been split into two, one for the lowest operating frequencies and another one for the highest frequencies. A good agreement between measurements and simulations is again observed. The designed antenna shows a higher directivity in the E-plane (the aperture plane), especially at higher frequencies. This is desirable to boost the detection capabilities in systems based on dielectric permittivity differences between the target and the surroundings.

Finally, the gain of the antenna is shown in Figure 6, both simulated and calculated from the measurements of the transmission parameter |S_21_| in the direction of maximum radiation. As shown, the results are again similar in both cases, obtaining a calculated gain between 2.0 dBi (low frequencies) and 6.5 dBi (highest frequency).

#### Broadband Antenna Time-Domain Analysis

In the medical imaging context, due to the accuracy and detection precision requirements, the signals involved are usually broadband. These signals are very short-time pulses including a broad frequency range, like the ones emitted by the proposed antennas. In this sense, these narrow pulses are highly prone to dispersion, which hinders the possibility of the reflected-back pulse in the antenna to be identical to the emitted one. To account for this phenomenon, the transmitted pulse is analysed in this section in the time domain in order to provide an estimation of the distortion likely to appear.

For the sake of simplicity, the analysis of the response is carried out in the frequency domain, and the results are further processed later for a suitable conversion to the time domain. In the proposed system, a pulse is emitted by one antenna, reflected onto a body within the measurement area, and received back by the same antenna. It should be noted that this scenario is essentially equivalent to a single transmission between a couple of face-to-face identical antennas separated by twice the distance between the single antenna and the body originating the reflection. In this equivalent scenario, the transfer function is straightforwardly obtained as the S_21_ parameter.

Therefore, two twin antennas have been placed one in front of another in the anechoic chamber, with a separation of 40 cm for ensuring far-field transmission. The S_21_ parameter has been recorded both from measurement and simulation, shown in Figure 7a. Vertical dotted lines mark the limits of the antenna bandwidth. It can be observed that there is a good agreement between simulation and measurement. The group delay, calculated from the phase derivative of the S_21_ parameter, has been also analysed and plotted in Figure 7b. It can be seen that it is fairly flat within the antenna bandwidth, showing values between 0.20 and 0.25 ns. These results suggest a low distortion in this system for signals whose bandwidth falls within the antenna bandwidth.

To confirm the previous result, an analytic time-domain analysis of a broadband pulse transmission can be done. The procedure starts from a Gaussian pulse given by:(1)$$G\left(t\right)=\frac{1}{\sqrt{2\pi }\sigma }\times e^{\left(-\frac{t^{2}}{2\sigma^{2}}\right)},$$
where *σ* is the time pulse width at half power. The transmitted broadband pulse used in this work is the fifth derivative of the Gaussian pulse, whose expression is the following:(2)$$T_{S}\left(t\right)=G^{5}\left(t\right)=\frac{d^{5}G}{dt^{5}}={\left(-1\right)}^{5}\times \frac{1}{{\left(\sqrt{2}\sigma \right)}^{2}}\times H_{5}\left(\frac{t}{\sqrt{2}\sigma }\right)\times G\left(t\right),$$
where *H*_5_(*t*) is 5th-order Hermite polynomial, defined in the following expression:(3)$$H_{5}\left(t\right)=32t^{5}-160t^{3}+120t.$$

The fifth derivative of the Gaussian pulse, *G*^5^(*t*), is used because its spectrum fits the frequency mask imposed by the VNA bandwidth (1 MHz–6 GHz). Figure 8a shows, in the time domain, the broadband pulse together with the original Gaussian pulse. The broadband pulse to be transmitted is defined in MATLAB (MATrix LABoratory), v. R2020a from Mathworks, Natick (Massachusetts, USA), and a Fast Fourier Transform (FFT) is done to obtain its frequency response. Figure 8b shows, in the frequency domain, the broadband pulse together with the VNA mask. This information can be then multiplied with the transfer function of the antenna system (Figure 7a) to obtain the received signal in the frequency domain. The received signal in the time domain, *R_S_*(*t*), can be obtained by applying the inverse FFT. The measured received signal can be obtained similarly by substituting the simulated transfer function with the measured one.

The time-domain received pulse is shown together with the transmitted pulse in Figure 9, using both simulated and measured function transfer for comparison. As can be seen, the received signal presents a qualitatively low distortion produced by the antennas. 

The best way to quantify this effect is to use the System Fidelity Factor (SFF), which is a measurement of the correlation between the transmitted and received pulses. This factor computes the relationship between the energy of the convolution between the transmitted and received pulses and the energy of each pulse separately [29]. The expression for calculating the SFF is:(4)$$\mathrm{SFF}=max_{n}\left|\frac{\int_{-\infty }^{+\infty }T_{s}\left(t\right)R_{s}\left(t+\tau \right)d\tau }{\sqrt{\int_{-\infty }^{+\infty }{\left|T_{s}\left(t\right)\right|}^{2}dt}\cdot \sqrt{\int_{-\infty }^{+\infty }{\left|R_{s}\left(t\right)\right|}^{2}dt}}\right|,$$
where *T_S_* is the transmitted pulse and *R_S_* is the received pulse. Therefore, SFF takes into consideration the distortion induced by the two antennas.

The SFF value obtained for this system is 95.29% when the simulated transfer function is used, and 96.97% if the measured S_21_ parameter is used, which indicates the high signal integrity of the signals transmitted by the antennas.

### 2.2. Medical Imaging Generation

In order to generate the medical images with the proposed antenna system, the signals reflected back to the antennas must be analysed and processed. These reflections are notably influenced by the permittivity differences between the healthy and malignant tissues inside the breast. These differences have an impact on the propagation and reflection of the electromagnetic waves. Therefore, the reflected signal captured by each antenna in the presence of malignant tissue is different depending on its position relative to the malignant tissue, and thanks to the joint analysis of the responses of the 16 antennas, the medical image can be processed. A depiction of the concept for the medical imaging system is shown in Figure 10. It should be noted that Figure 10 is limited to a schematic representation of the system. In the real system, the antennas have a different orientation, perpendicular to the positions shown in Figure 10 (as shown in Figure 1).

In this sense, in addition to the hardware system, the image-generation system (also referred to as image-formation algorithms or image-building algorithms) is crucial due to its decisive influence on the overall performance of the system. Image-generation systems based on broadband antenna measurements are currently facing several challenges in regards to breast cancer detection. On the one hand, active research is constantly being conducted on the optimal design of the antennas and the production of realistic breast phantoms to facilitate experimental study, development and validation. On the other hand, image-formation algorithms must be mastered, since they should provide considerably high detection capabilities, accurate location of the key points or elements, great robustness and fast computational speed.

Many medical image-formation algorithms have been proposed during the last several years. Li and Hagness [30] proposed for the first time the so-called Confocal Microwave Imaging (CMI) technique, which generates the image by using a method based on Delay-and-Sum (DAS) algorithms. From these principles, a number of enhanced or adapted algorithms have been proposed, which are usually divided into two main groups: Data-Dependent (DD) and Data-Independent (DI) ones. DD algorithms rely on the premise that the conditions of the system under measurement (breast) are known. This condition is difficult to be met in real scenarios, and this is why these algorithms find quite restricted applications. DI algorithms, however, do not require this prior information to produce an accurate image of the system. Nowadays it is easy to find a considerable amount of DI algorithms such as Delay-Multiply-and-Sum (DMAS) [31] and Improved Delay-and-Sum (IDAS) [32], among others.

In this proposal, as described above, the system is fully operated by a master computer, which handles the breast measurement process. Figure 11 depicts the full flow diagram of the system, which is prepared to work autonomously. Automation is reached thanks to the above-mentioned Python script, which keeps in simultaneous communication with the microcontroller (which masters the switching network to select the corresponding active antenna) and with the VNA, as shown in Figure 1, and masters the measurement process. The reflection coefficient (magnitude and phase) seen in each antenna is measured twice: first, when the system is empty, to use this measurement as a reference, and second, with the breast or model under analysis.

After gathering the reflection parameters for all the antennas, the signal processing is done to build the resulting medical image. Before running the image-building algorithm, a preprocessing stage is applied in order to eliminate noise and unwanted artefacts. Most of the noise is eliminated thanks to applying a Hamming window to the samples, provided that the interesting information is generally found in the middle of the measurement spectrum. Then, the inverse chirp Z-transform is applied both to the reference measurement and to the breast/model measurement. We selected the inverse chirp Z-transform because it features a good reconstruction of the signals in the time domain. The input broadband pulse must also be eliminated, so that only information relative to the reflections is present. To that end, the reference measurement (χ_system_) is subtracted from the breast/model measurement (χ_breast_).

It is also conducive to our interests to eliminate the reflections due to the outer skin surrounding the breast, so that a better definition of the tumour is achieved. Outer breast skin has a usual depth between 1 and 3 mm, and it shows significantly higher permittivity values than those of the inner breast tissues [23]. This yields considerable reflections due to this outer skin, which may mask the reflections associated with the tumour tissues. To avoid this, we average out the signals from all the antennas, the result of which is used as a calibration signal, following the method described in [33]. This calibration signal is later subtracted from each antenna response (*χ*) as shown in the following expression:(5)$$\beta_{m}\left(n\right)=\chi_{m}\left(n\right)-\frac{1}{M}\sum_{j=1}^{M}\chi_{j}\left(n\right),$$
where *β**_m_* are the time-domain signals for each antenna, *M* is the number of antennas (16 in our case) and *n* is the sample index. This method is suitable for cases where the antennas are all at the same distance from the breast model, and where breast models with highly uniform distribution are involved, as in this case. This step is referred to as breast-skin artefact removal in the flowchart of Figure 11.

After this preprocessing, the signals are ready for medical image-building processing. To do it, we propose an IDAS algorithm [32], which makes use of a DAS algorithm [30] and includes a new weighting vector called coherence factor (*CF*). A DAS algorithm starts from a point *r*_0_ inside the breast, and it computes the delay between the position of each antenna and *r*_0_. This is done thanks to an estimation of the dielectric properties of the breast at the centre frequency of the broadband pulse. These delays, defined in (6), allow the system to isolate the response of each antenna for each point *r*_0_. Finally, the integral of the quadratic sum of the intensities for all the responses during a defined time window is calculated. The expression for the delays is:(6)$$\tau_{m}\left(r_{0}\right)=\frac{d_{m}\cdot \sqrt{\varepsilon_{r}}}{c}\times f_{s},$$

With *d_m_* being the distance between each antenna and the point *r*_0_, *f_s_* the sampling frequency, *c* the speed of light in vacuum and *ε*_r_ the relative permittivity of the breast tissue, estimated at the centre frequency of the broadband pulse.

In an IDAS algorithm, furthermore, an additional parameter, *CF*, is involved, which includes a measurement of the coherence of the broadband signals reflected inside the breast for a certain point *r*_0_. A high coherence of the received signals implies the presence of a tumour inside the breast. The *CF* parameter is computed as:(7)$$CF\left(r_{0}\right)=\frac{{\left[\sum_{n=1}^{M}\beta_{n}\left(\tau_{m}\left(r_{0}\right)\right)\right]}^{2}}{\sum_{n=1}^{M}{\left|\beta_{n}\left(\tau_{m}\left(r_{0}\right)\right)\right|}^{2}}.$$

The parameter *CF* is implemented into the IDAS equation in the following manner:(8)$$I\left(r_{0}\right)=CF\left(r_{0}\right)\int_{0}^{T_{Win}}{\left[\sum_{n=1}^{M}\beta_{n}\left(\tau_{n}\left(r_{0}\right)\right)\right]}^{2}dt,$$
where β*_n_* are the radar signals depending on the delay τ*_m_* with respect to *r*_0_, which are the different locations of the points inside the objects or models to analyse.

## 3. Calibration and Start-Up of the System

After describing the system for early detection of breast tumours, in this section the prior calibration and adjustments for the suitable production of the medical images and results are shown. Initially, a complete calibration of the measurement system must be done to properly eliminate the delays and reflections of the signals generated by the VNA and transmitted by each antenna. A full SOL (Short–Open–Load) calibration, carried out with the Rodhe & Schwarz ZV-Z135 calibration kit, is applied to each antenna when the system is ready, taking as the calibration reference plane the SMA connector of each antenna. Making this calibration when the system is ready and all the elements and cables are in the same positions as they will be during the measurements provides for the proper correction of all the possible errors in the system. This is equivalent to having the VNA directly connected to each antenna. Using this procedure, 16 calibration files (one per antenna) are generated and saved in the VNA. During the measurements, the VNA loads sequentially the corresponding calibration file to perform the measurement with the corresponding antenna.

Once the system is calibrated as described before, an empty reference measurement (without any kind of body or object inside the system) is made in order to take into account the reflections and possible effects of the system itself. This reference measurement will be used later in all the experimental measurements for eliminating the system reflections and maximising the sensitivity to the reflections caused by the phantoms or objects under measurement.

Some extra initial trials with simple elements have been performed with the purpose of fine-tuning the system and its detection capabilities for the position and dimensions of the phantoms under analysis. These elements are of known size and they provide for notable, clear reflections of the electromagnetic waves. Figure 12a shows a picture of one of them: a solid metal cylinder with a diameter of 5.0 cm and a height of 11.5 cm, placed exactly in the centre of the measurement area. This means that all the antennas should detect the object at the same distance. Figure 12b plots the time-domain received signal in each antenna when the system is empty (no objects inside), which is used as reference measurement, whilst Figure 12c plots the time-domain received signal in each antenna when the cylinder is placed in the centre of the measurement area. It can be seen that the obtained responses are quite similar for all the antennas (as logical since they are all at the same distance from the cylinder), and there is also a certain difference with respect to the reference measurement.

Further processing the measurements, Figure 12d shows the time-domain signals after subtracting the reference measurement signal from the cylinder measurement signal. It can be seen that all the antennas show a remarkable reflection at the same time, which has been adjusted to the exact position of the cylinder. This position adjustment must be done while fine-tuning the system because the time pulses are generated mathematically, which prevents them from being duly adjusted to their coordinates origin. When the time axis (and therefore the distance detection) has been adjusted thanks to the cylinder object, which in practice means an axis translation, the adjustment is applied to the rest of measurements. Figure 12e shows the reflection intensity in the response of each antenna after the subtraction and when the Hamming window has been applied. It can be observed that the noise is reduced, and all the antennas achieve a clear detection of the cylinder at the same distance. Once the object detection distances for all the antennas have been adjusted, the image is generated by means of an IDAS algorithm, as detailed in the prior section. This provides for a two-dimensional representation of the cylinder under analysis, as can be seen in Figure 12f. This final image shows a clear reflection due to the cylinder at the appropriate distance from all the antennas and with reference to the coordinate system of the proposed setup.

After checking that the performance of the system is appropriate when the simple metal cylinder object is involved, a further checking-and-adjustment process is carried out with a more complex object, similar to the phantoms that will be used later to assess the breast tumour detection capabilities of the system. For this new trial the object under analysis is a 3-D-printed plastic container with hemispherical shape. It has been made with standard polylactic acid (PLA), and it has a fillable inner volume of 55.7 dL. Figure 13 shows a picture of the object inside the measurement area.

The container has been filled with a basic mixture of simple materials (sunflower oil, margarine and wheat flour). This mixture has a dielectric constant and absorption consistent with those of breast adipose tissue [34,35]. The dielectric properties of the mixture have been measured with a simple resonator cell at 2.5 GHz, resulting in *ε*_r_ = 6 and tan δ = 0.34. Also, acting as simple models for a tumour, some fillable plastic cubes have been 3-Dprinted with different inner volumes ranging from 0.2 to 2.0 mL (Figure 13 bottom). The cubes can be filled with salty water solutions, which shows a considerably high dielectric constant and loss tangent (*ε*_r_ = 72 and tan δ = 0.55 at 2.5 GHz). The considerable contrast between the dielectric properties of the tumour-mimicking cubes and those of the breast-adipose-tissue-mimicking container makes their detection and, consequently, the calibration and fine-tuning process, easier.

For instance, Figure 14 shows the images obtained from the complete processing of the signals obtained when the 0.2 mL tumour-mimicking cube is placed next to antenna #13 (position at 270°). Figure 14a plots the time-domain responses of the antennas after subtracting the reference measurement from the phantom measurement when the breast-skin artefact removal is not applied, and Figure 14b shows the resulting IDAS image. All the information of the objects inside the measurement area is involved in this image. However, being an intensity plot, the high-intensity reflection in the boundary of the container masks the rest of the information. To eliminate the masking effect of the first reflection, we applied average-signal-extraction processing, the breast-skin artefact removal, which eliminates the effect of this first reflection in the final image. This renders the inner reflections more noticeable, as shown in Figure 14c,d. It can be seen how the effect of the first reflection disappears from the responses of the antennas (Figure 14c) and the information of the rest of the objects becomes easily noticeable (Figure 14d), the tumour phantom involved in this case being thereby easily detectable.

It is interesting to note that the whole process implies exposure times no longer than 1 min, and the waves in the proposed working frequency range are non-ionising. Furthermore, the antennas emit a power of 0 dBm = 1 mW, which is between 10 times and 100 times lower than the usual emission power in a modern smartphone. This power is also two orders of magnitude lower than the threshold power required to reach the maximum allowed specific absorption rate (SAR) according to the standards considered in US and European Union regulations [36]. All these reasons allow the system to be deemed suitable for use in a safe manner in regards to human exposure and SAR requirements.

## 4. Results and Discussion

Once the system is ready to use, and having checked its capabilities for detecting strange elements inside simple biological phantoms, we further assessed the tumour detection capabilities of the system when more realistic breast phantoms are concerned. We used phantoms with dielectric properties closely resembling those of real breast tissues. Also, we involved different tumour phantoms with different electric properties, mimicking the different sorts of tumours to be usually found. Furthermore, the phantoms have been prepared so that the difference between the properties of the breast phantom and those of the tumour phantom becomes smaller for each new trial, thereby hindering the detection process with the proposed system. The results of the experimental procedure are shown and discussed in the next subsections.

### 4.1. Breast and Tumour Phantoms

The product TRITON X-100 was used for the design of the breast and tumour phantoms. It is a soapy chemical compound widely used in the scientific literature mostly due to its stability when achieving biological phantoms with different electric properties [23,37,38]. The chief advantage of TRITON X-100 is that it dissolves easily in water, and by properly modifying the proportions of TRITON X-100 and water it is possible to attain a wide range of electrical properties for the final compound.

With the purpose of evaluating the possibilities of TRITON X-100, several mixtures with distilled water were prepared and their electrical properties measured. To do this, we used the common dielectric-characterisation method based on a coaxial probe [39,40], with a probe and a software script devoted to the calculation of the dielectric constant (*ε*_r_) and the conductivity (*σ*), both developed by our team. Figure 15 shows the results for both parameters in the 1 MHz–6 GHz frequency range, sorted out according to the concentrations of the solutions (ranging from pure distilled water to pure TRITON X-100 in steps of 10% volume concentration).

Figure 15 shows that a remarkably wide range of dielectric constant and conductivity values can be attained with the proper concentration of TRITON X-100. This can be used to create biological phantoms for a large variety of tissues. In this case, we have selected a TRITON X-100 concentration of 50% for the breast phantom, which yields a phantom with similar characteristics to real breast tissue [23]. In addition, we have chosen several concentrations for the tumour phantoms so that the detection capabilities of the proposed microwave system can be assessed for different cases involving different contrasts of microwave signal absorption in reference to the surrounding medium [41]. A summary of the characteristics of the different biological phantoms involved is shown in Table 2. It should be noted that tumour phantom T4 shows a remarkably low dielectric contrast with respect to the breast phantom, thereby allowing us to address the early-tumour-detection challenge, as well as the dense-breast cancer-screening one [21,24].

Studies involving in vivo measurements in a considerable number of individuals have shown an average dielectric constant for medium-density breasts of approximately 24.5 ± 8.0 and a conductivity of approximately 2.8 ± 0.6 at 3 GHz [43,44]. As shown in Table 2, the breast phantom used here matches the electromagnetic absorption and scattering properties of real tissue in breasts with a certain density, thus emulating the tumour detection in dense breasts. Considering the dielectric constant of the breast phantom and the maximum frequency in the system, the spatial resolution is 5 mm.

### 4.2. Detection of Different Tumour Phantoms

For assessing the tumour-detection capabilities of the proposed system and algorithms, an anatomic breast phantom was 3-D printed with PLA (*ε*_r_ = 2.88, tan δ = 0.02 [45]) and filled with 55.7 dL of the breast TRITON X-100 mixture (see Table 2). As tumour phantoms, we used small rounded 3-D printed containers with inner volumes of 1 mL and 2 mL, filled with the TRITON X-100 mixtures shown in Table 2. It should be noted that the PLA piece acts only as a container for the breast phantom, and it has no effect on the final images thanks to the breast-skin artefact removal algorithm. Figure 16 shows some pictures of the breast phantom, its morphology, and its position inside the measurement area. One of the tumour phantoms (T1, 2 mL) and its position inside the breast phantom can also be seen.

The method described in [46,47] has been applied to obtain an estimation for the signal-to-noise ratio (SNR) of the system. The noise of the system was computed by means of measuring the OPEN calibration standard at the position of one of the antennas, yielding a noise level always lower than −55 dBm in the whole frequency band. A measurement was made with the corresponding antenna, with 0 dBm emission power, involving the breast phantom including a 1 mL T1-kind tumour, yielding a signal level always greater than −30 dBm in the whole frequency band. Therefore, for the proposed system it can be assumed that SNR ≥ +25 dB for the whole frequency band.

Some experiments have been done with the proposed phantoms and setup in order to evaluate the tumour-detection capabilities of the system. The main results are shown and discussed in the following subsections.

#### 4.2.1. High-Electromagnetic-Absorption Tumour Phantoms

In a first approach, we performed measurements and analysis in two situations involving tumour phantoms with high electromagnetic absorption, high dielectric constant and high conductivity, with 1 mL and 2 mL inner volume. We used the T1 tumour phantom (Table 2), which was placed at 2 cm from the breast phantom boundary at a 90° position (close to antenna #5).

The resulting IDAS images for these measurements are shown in Figure 17. It can be seen that the system is fully capable of successfully detecting the tumour phantom in spite of the remarkable reflection in the outer border of the breast phantom. In both cases, a clear detection of the tumour is seen close to antenna #5, which also allows for accurate location. The border-removal algorithm, nevertheless, is not perfect, and it can be seen in these images that a small error of this algorithm appears at position ∼0° (close to antenna #1) located in the phantom boundary area. This error could also be due to the fact that the breast phantom was not perfectly centred in the measuring area, which is a requirement for accurate border removal. All in all, this small artefact shows a considerably lower intensity than the tumour phantom, and it does not hinder the proper detection of the tumour. Having checked the performance and potential possibilities of the proposed system, further experimental assessment will be shown in the next subsections.

#### 4.2.2. Moderate-Electromagnetic-Absorption Tumour Phantoms

After showing that the system is capable of detecting tumours with high electromagnetic absorption (which applies to tumoural cells with considerably high water content and highly virulent tumours), some experiments were made with lower absorption values in the tumour phantoms. To that end, the tumour phantoms T2, T3 and T4 (Table 2) are been considered. The dielectric constant and conductivity values of these phantoms are associated with less-aggressive tumours, as well as early-stage tumours. In this case the detection process becomes more complicated since the contrast between the electric characteristics of the tumour phantoms and those of the breast phantom becomes less noticeable.

The results for the measurements with two T2-kind tumour phantoms with inner volumes 1 mL and 2 mL, placed in the same position as before, can be seen in Figure 18. It can be observed that, again, the final image clearly shows the existence of the tumour and its position close to antenna #5 (90°) for both cases, despite the lower dielectric contrast between the tumour and breast phantoms.

Moreover, two 1 mL tumour phantoms of T3 and T4 kinds have been used in the same position as before. These are the phantoms showing the lowest electromagnetic wave absorption and the lowest volume, i.e., the most difficult ones to detect. Figure 19 shows the results of these measurements. As can be seen, the system properly detects the existence and position of the tumours in both cases. The reduced dielectric contrast between the breast and the tumour phantoms leads to new slightly illuminated areas in the images, especially for T4 (Figure 19b), but the proper identification of the tumour is still clear and unambiguous. 

This result demonstrates the tumour-detection capabilities of the proposed system even for cases of low difference between the properties of the breast phantom and those of the tumour phantoms. The improvement of the detection capabilities which enables the system to achieve this low-dielectric-contrast detection can be ostensibly attributed to two main features of the proposed system: (i) highly-directive antennas in the E-plane, which allows for accurate detection of dielectric constant differences; and (ii) the use of an improved image-generation algorithm (IDAS) with precise calibration and artefact removal preprocessing. This result allows us to envision the potentialities of the proposed system for early cancer detection and for facing cancer detection challenges in dense breasts and/or young subjects.

#### 4.2.3. Multiple Tumour Detection

Finally, additional measurements have been made involving several tumour phantoms inside the breast phantom at the same time. Two T1-kind tumour phantoms with inner volumes of 2 mL and 1 mL have been placed at 0° and 180°, close to antennas #1 and #9, respectively. The resulting image for this new setup is shown in Figure 20, where both tumour phantoms are clearly visible (they have been highlighted with two coloured circles: a red circle for T1 2 mL and a white circle for T1 1 mL). As is logical, the information associated with the tumour phantom with inner volume of 2 mL is more accentuated than that of the tumour phantom with inner volume of 1 mL, since the reflection due to the biggest tumour phantom is more remarkable, especially in the case of a high dielectric constant, as happens with T1.

This fact notwithstanding, both tumour phantoms can be clearly identified and distinguished, which demonstrates the detection robustness and accuracy of the proposed system. An enhanced precision could be achieved with a larger number of receiving antennas active at the same time for each transmitted pulse, at the cost of a more expensive system, a higher complexity and longer processing times. As a trade-off, for each transmitting antenna, in future works we will investigate the use of the neighbour antennas as well as the antenna in front of it as receiving antennas. The extra information from the neighbour antennas could contribute to increasing the precision of the detection and location of tumours, whereas that from the antenna in front of the emitting one could be helpful to achieve a tailored estimation of the permittivity for each breast, which would allow for better spatial location of the tumours.

## 5. Conclusions

This work shows a non-invasive microwave-based breast cancer detection system. All the parts and components have been designed with the aim of achieving the most autonomous and portable capabilities possible. With this purpose, the entire hardware system has been designed and developed with high-frequency passive elements, including a microcontroller, a VNA for signal measurement and a computer for the automation and control of the process. The software system involves a Python script which implements the switching control and measurement processes so that the data can be obtained in an automatic way. It also includes the preprocessing and the medical imaging algorithms.

Calibration measurements with simple and known models, such as a metal cylinder, have been performed in order to evaluate the correct functioning of the system and adjust the algorithms. For the experimental assessment, breast tumour tissue has been emulated by suitable mixtures of water and TRITON X-100, a synthetic material often used for such a purpose. The changes in the dielectric constant of these mixtures are characterised depending on the mixing ratio, so that suitable models can be obtained. Finally, a breast phantom is considered for the experimental validation made with 50% TRITON X 100 mixture, and four tumour phantoms (at four different concentrations, showing four different dielectric constant contrasts) are involved.

The results and images obtained show the capability of the system to detect and locate the different sorts of tumour phantom, including phantom volumes as small as 1 mL, even for low dielectric constant contrast and even when more than one tumour is present at the same time. The use of highly directive antennas and an improved medical imaging algorithm (IDAS), in addition to an adaptive preprocessing aimed at artefact and undesired reflection removal allows for effective detection of all the tumour phantoms considered here. The results reported in this study show the potential capabilities of the proposed system to address the current challenges involved in early breast cancer detection and cancer detection in dense breasts.

## Figures and Tables

**Figure 1 biosensors-12-00752-f001:**
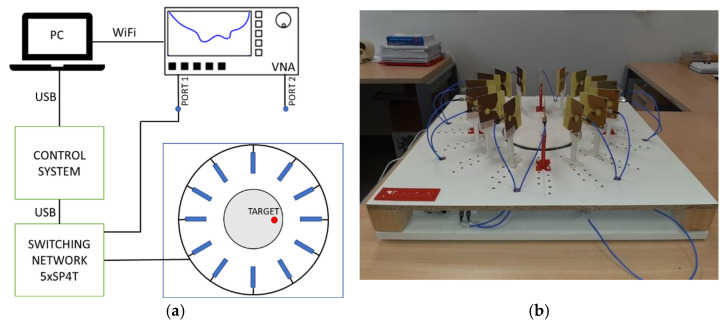
Proposed microwave system for early detection of breast tumours: (**a**) Scheme of the system; (**b**) Picture of the measurement scenario.

**Figure 2 biosensors-12-00752-f002:**
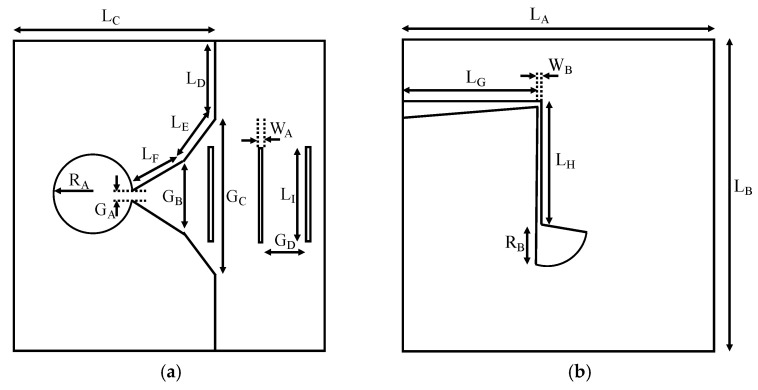
Antenna scheme and dimensions: (**a**) Top face; (**b**) Bottom face.

**Figure 3 biosensors-12-00752-f003:**
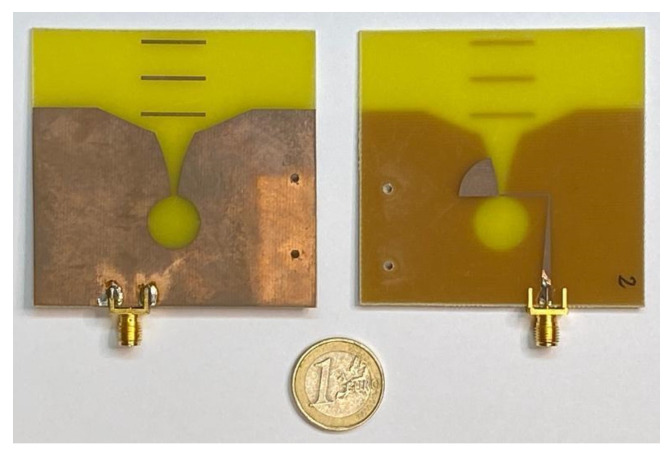
Implementation of the improved Vivaldi antenna proposed in this work.

**Figure 4 biosensors-12-00752-f004:**
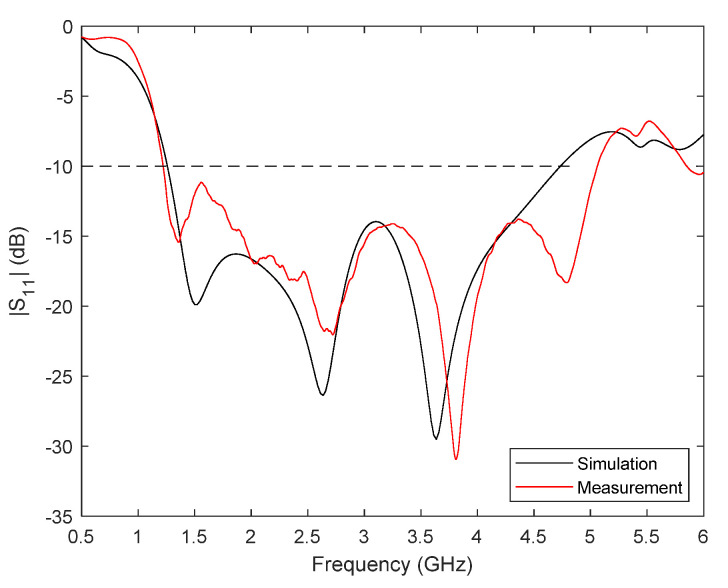
Measured and simulated return losses |S_11_| for the modified Vivaldi-like antenna.

**Figure 5 biosensors-12-00752-f005:**
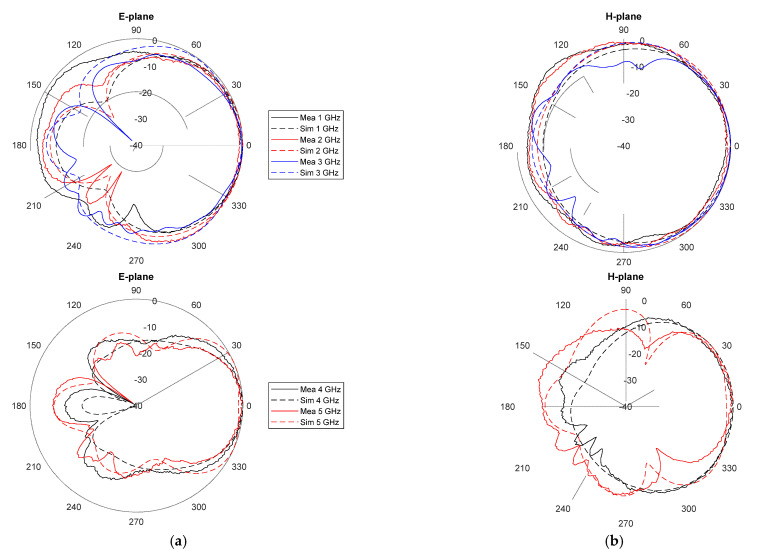
Measured and simulated radiation patterns of the proposed antenna for several frequencies within the operating range (top: lowest frequencies; bottom: highest frequencies): (**a**) E-plane; (**b**) H-plane.

**Figure 6 biosensors-12-00752-f006:**
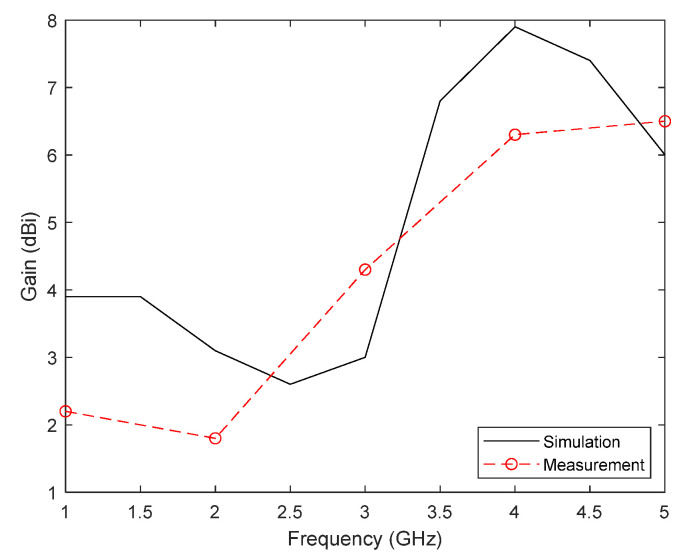
Measured and simulated gain of the proposed antenna.

**Figure 7 biosensors-12-00752-f007:**
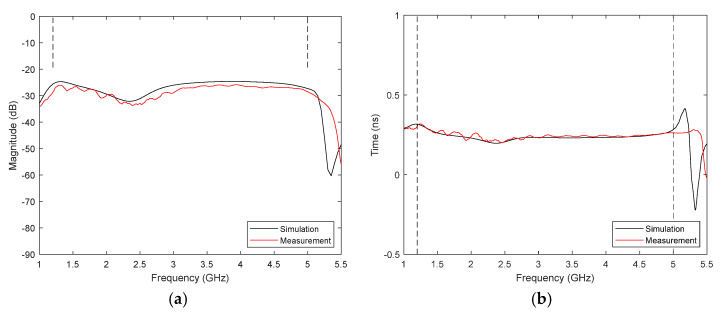
Transfer function of the proposed transmission scheme: (**a**) Module; (**b**) Group delay.

**Figure 8 biosensors-12-00752-f008:**
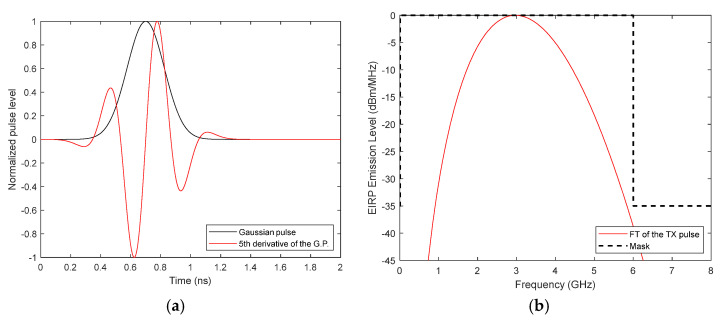
Plots of the Gaussian pulse: (**a**) Time-domain comparison between the Gaussian pulse and the fifth derivative of the Gaussian pulse; (**b**) Frequency-domain representation of the fifth derivative Gaussian pulse and VNA mask.

**Figure 9 biosensors-12-00752-f009:**
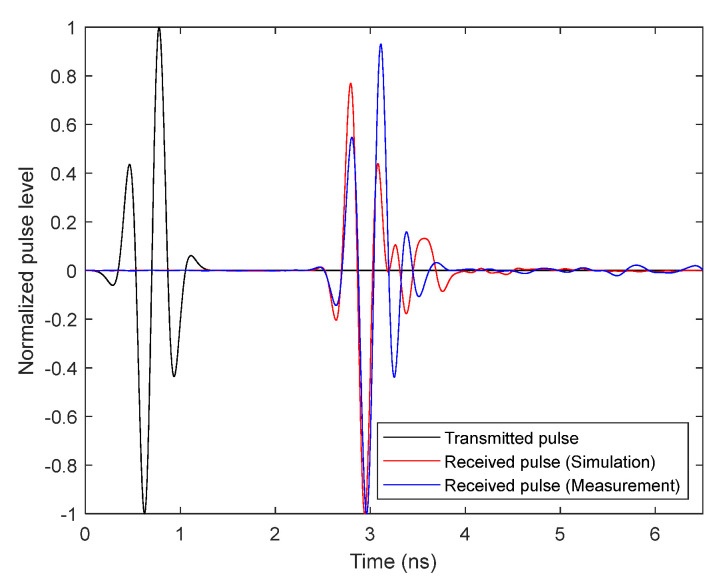
The transmitted pulse compared with the received pulse when simulated and measured transfer functions are used.

**Figure 10 biosensors-12-00752-f010:**
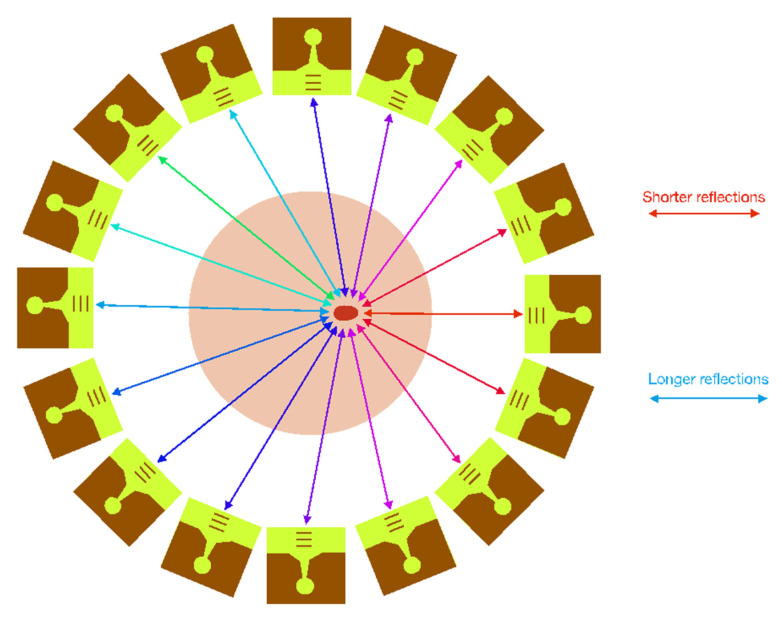
Concept of the proposed medical imaging system.

**Figure 11 biosensors-12-00752-f011:**
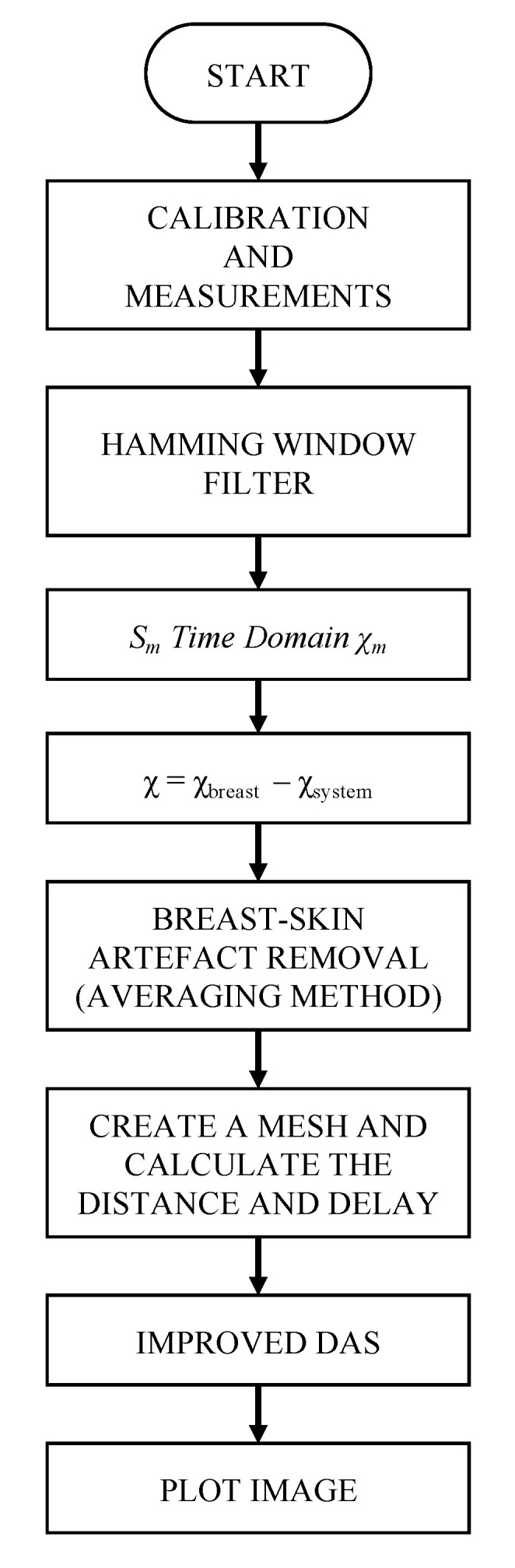
Flowchart of the digital data processing.

**Figure 12 biosensors-12-00752-f012:**
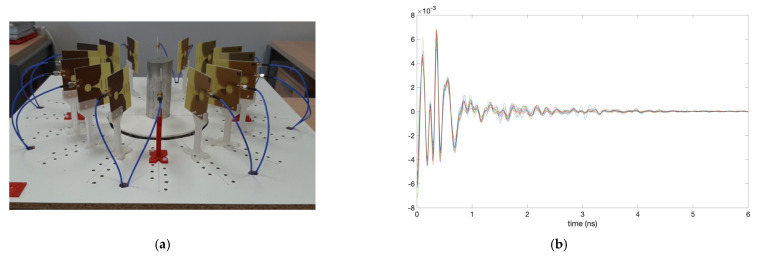
Fine-tuning process involving a metal cylinder of known dimensions: (**a**) Picture of the cylinder placed in the centre of the antenna system; (**b**) Time-domain received signal in each antenna for the reference measurement (without the cylinder); (**c**) Time-domain received signal in each antenna for the measurement with the cylinder; (**d**) Time-domain signals after subtracting the reference measurement from the cylinder measurement; (**e**) Reflection intensities in the responses of the antennas after subtracting the reference measurement and applying Hamming window; (**f**) Final generated image with IDAS algorithm, including the position of the antennas.

**Figure 13 biosensors-12-00752-f013:**
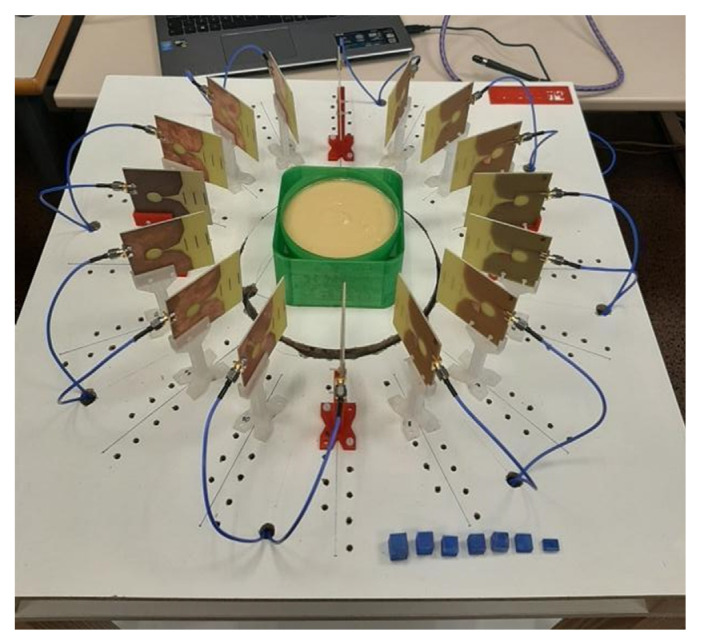
Picture of the 3-D-printed object used for fine-tuning the system.

**Figure 14 biosensors-12-00752-f014:**
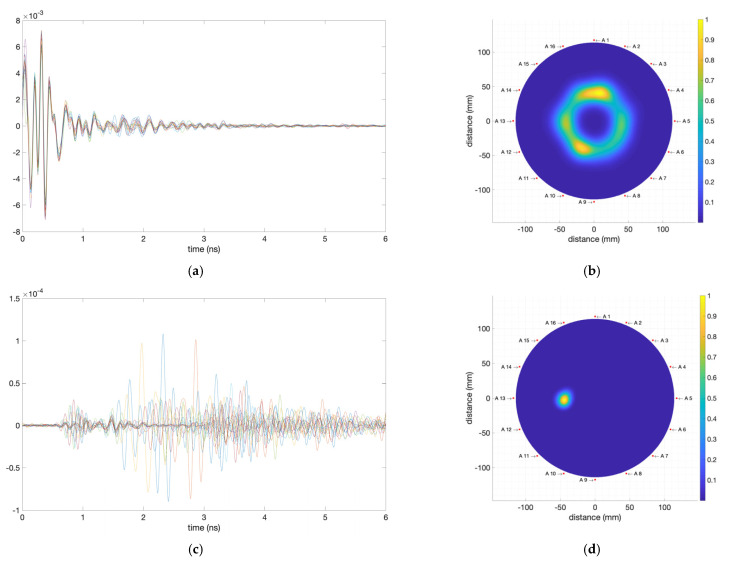
Fine-tuning process involving the breast-adipose-tissue-mimicking and the tumour-mimicking phantoms: (**a**) Time-domain responses of the antennas after subtracting the reference measurements from the phantom measurements without applying the breast-skin artefact removal; (**b**) IDAS image without applying the breast-skin artefact removal; (**c**) Time-domain responses of the antennas after subtracting the reference measurements from the phantom measurements and applying the breast-skin artefact removal; (**d**) IDAS image when the breast-skin artefact removal is applied.

**Figure 15 biosensors-12-00752-f015:**
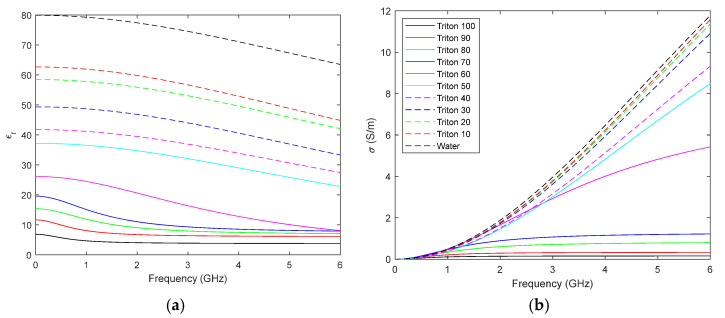
Characterisation of different mixtures of TRITON X-100 and distilled water: (**a**) Dielectric constant; (**b**) Conductivity.

**Figure 16 biosensors-12-00752-f016:**
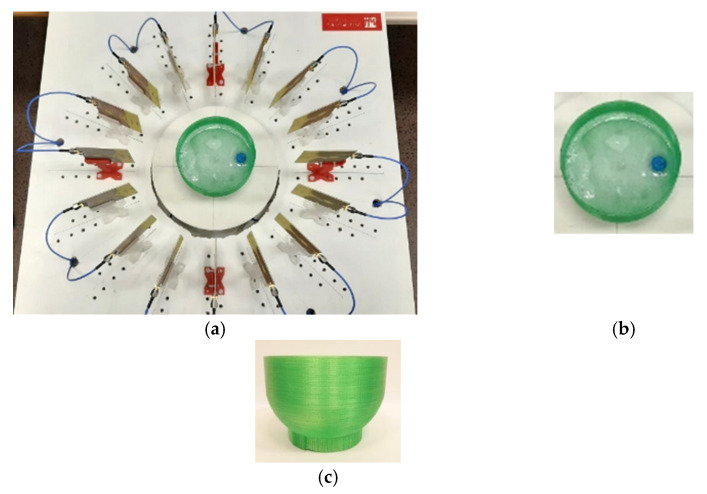
Breast and tumour (T1, 2 mL) phantoms used in the experimental tests: (**a**) Measurement setup; (**b**) Upper view of the breast–tumour phantom; (**c**) Side view of the breast phantom.

**Figure 17 biosensors-12-00752-f017:**
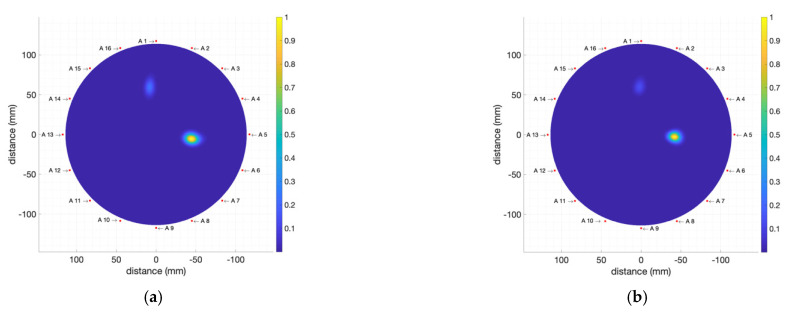
Obtained images (including breast-skin artefact removal) for the breast phantom including: (**a**) T1 2 mL tumour phantom; (**b**) T1 1 mL tumour phantom.

**Figure 18 biosensors-12-00752-f018:**
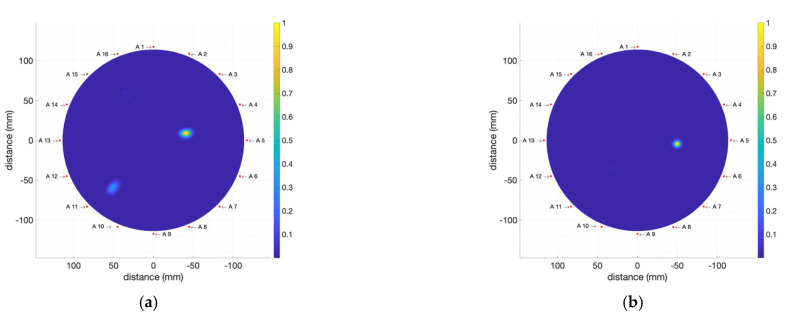
Obtained images (including breast-skin artefact removal) for the breast phantom including: (**a**) T2 2 mL tumour phantom; (**b**) T2 1 mL tumour phantom.

**Figure 19 biosensors-12-00752-f019:**
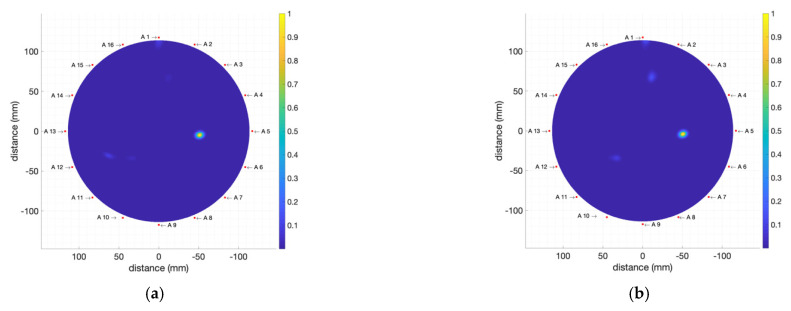
Obtained images (including breast-skin artefact removal) for the breast phantom including: (**a**) T3 1 mL tumour phantom; (**b**) T4 1 mL tumour phantom.

**Figure 20 biosensors-12-00752-f020:**
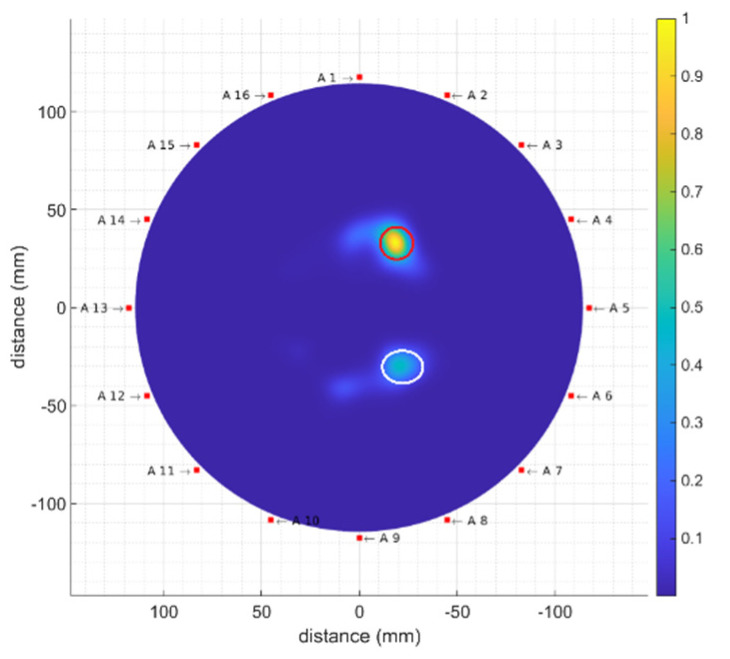
Obtained image (including breast-skin artefact removal) for the breast phantom including T1 1 mL and T1 2 mL phantoms at the same time. The detected tumours have been highlighted with two coloured circles: red circle (T1 2 mL) and white circle (T1 1 mL).

**Table 1 biosensors-12-00752-t001:** Final dimensions of the designed antenna.

Dimension	Value	Dimension	Value	Dimension	Value
L_A_	68.0 mm	L_G_	27.7 mm	G_B_	9.9 mm
L_B_	70.0 mm	L_H_	12.8 mm	G_C_	38.0 mm
L_C_	49.0 mm	L_I_	16.0 mm	G_D_	8.0 mm
L_D_	16.0 mm	W_A_	1.0 mm	R_A_	7.0 mm
L_E_	15.2 mm	W_B_	0.6 mm	R_B_	9.4 mm
L_F_	34.2 mm	G_A_	1.3 mm		

**Table 2 biosensors-12-00752-t002:** Breast and tumour phantoms involved.

Phantom	% TRITON X-100	% Distilled Water	% Seawater	*ε*_r_ @ 3 GHz	*σ* [S/m] @ 3 GHz
Breast	50	50	0	32	3.02
Tumour 1 (T1)	0	0	100	70	7.09 [42]
Tumour 2 (T2)	10	90	0	57	3.79
Tumour 3 (T3)	20	80	0	53	3.75
Tumour 4 (T4)	30	70	0	44	3.63

## Data Availability

Not applicable.

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
