# Peer review of "Non-Invasive Microwave-Based Imaging System for Early Detection of Breast Tumours"

_biosensors, 2022, doi:10.3390/bios12090752_

Round 1

Reviewer 1 Report

The manuscript proposed a microwave antenna array to detect breast cancer. The imaging system, data processing and performance analysis are presented in great details. However, the phantom study is not solid. Authors need to improve the experiment design to make the work publishable.

1.       “The pulse is reflected onto the different materials making up the objects or bodies under analysis, and the reflected signal is finally received (and recorded) by the active antenna, the same one that emitted it.”

Why is the same active antenna transmit and receive pulse instead of all antennas receiving pulse? Both CT, array ultrasound and photoacoustic tomography use multiple arrays to receive signals and the reconstructed image shows superior resolution and signal-to-noise ratio.

2.       What is resolution and imaging depth of the system?

3.       The breast phantom is printed with PLA, which forms a rigid shell. This is way different from a real tissue and thus make the result inconvincible.

4.       The antenna array proposed here is sensitive to electromagnetic absorption and scattering contrast of targets. Does the phantom used in the experiment match real tissue in terms of electromagnetic absorption and scattering?

5.       Clinically, does breast cancer have different electromagnetic properties in micro-wavelength from normal tissue?

6.       Clinically, does malignant tumor have different  electromagnetic properties in micro-wavelength from benign tumor?

7.       In introduction section, authors only mentioned drawbacks of X-ray mammography. However, more imaging modalities have been investigated for breast cancer detection, including photoacoustic tomography, diffuse optical tomography. Both have shown good penetration depth. Compared to other imaging method, what is advantage of  the microwave based imaging?

8.       Over traditional clinical imaging, including ultrasound, CT, MRI, PET, what is advantage of  the microwave based imaging?

Reviewer 2 Report

The authors have shown that a non-invasive microwave-based breast cancer detection system is viable. Calibration measurements with simple and known models, such as a metal cylinder, were performed to evaluate the right functioning of the system and train the algorithms. Breast phantom studies are creative and reliable. The results and images obtained showed capability of the system to detect phantom volumes of approx 1 mL, in presence of multiple tumour.  The article is well written and the field is of high impact. I recommend the publication of this article in Biosensors and hope that the audience will find it very interesting.

Reviewer 3 Report

This work fabricated a non-invasive microwave medical imaging system based on radar technology, which composed of hardware system and software system. This method exhibited promising tumours detection capabilities with high response speed and accuracy of key points or elements.

Comment 1: How would the morphology, size and thickness of breast tumours impact the imaging system?

Comment 2: Compared with commercial imaging system for early detection of breast tumours, what are the advantages of this method?

Comment 3: How about the signal-to-noise ratio of this imaging system?

Comment 4: The image formats need to be further unified, such as the frames in Figure 9 and Figure 8a.

Comment 5: The English expression and grammar should be further improved. 

Reviewer 4 Report

The manuscript under consideration presents a radar-based microwave imaging system for breast cancer detection. The authors proposed both hardware and software components, properly calibrated and tested them on the breast phantoms with different dielectric properties. The results show promising efficiency in detection of small-volume tumor phantoms. Proposed system could be a worth candidate for near future breast tumor imaging and diagnostics.

As a comment, I would suggest the authors to put an estimate of specific absorption rate on the phantom to prove that the antenna is safe for human exposure.

The manuscript can be accepted for publication with minor corrections.

Round 2

Reviewer 1 Report

Authors have addressed all questions well. In the response letter, authors pointed out that

'Quality (PAT) images are only obtained for certain sizes of targets due to the intrinsic limitations of ultrasound transducers. Targets larger than 6 mm are difficult to detect'.

This is incorrect, authors may read more latest related work. PAT technology is classified as PA computational tomography and PA microscopy. For the later one, it's ture 6mm is kind of limitation but for the first one, a few centimeters imaging depth is very feasible.

Overall the work is intersting and I recommend publishing the work.